

**Atmospheric CO₂ retrieval from ground based FTIR spectrometer over Shadnagar, India.**
Pathakoti Mahesh*, Gaddamidi Sreenivas, Pamaraju Venkata Narasimha Rao and Vinay Kumar
Dadhwal
National Remote Sensing Center (NRSC)
Indian Space Research Organization (ISRO)
Hyderbad-500037
Email-ids: mahi952@gmail.com; sreenivasg26@gmail.com; rao_pvn@nrsc.gov.in;
dadhwalvk@hotmail.com
*Correspondence to: Pathakoti Mahesh (mahi952@gmail.com)
























**Abstract**
The column-averaged volume mixing ratio (vmr) of carbon dioxide ($XCO_2$) has been retrieved
over Shadnagar (Latitude=17.03° N, Longitude=78.21° E), India at Near Infrared (NIR) spectra
using ground based high resolution ($\Delta v$=0.01 cm$^{-1}$) Fourier Transform InfraRed (FTIR; model
IFS125M) spectrometer. It is first of its kind in India for measuring columnar and vertical mixing
ratios of atmospheric trace gases and other greenhouse gases (GHGs). In the present study,
retrieval of $XCO_2$ could be performed for a week at near Sun's nadir view (solar zenith angle-
SZA, ±75.0°) using line-by-line radiative transfer algorithm (LBLRTA). Vertical profiles of $CO_2$
have been retrieved at NIR and middle IR (MIR) spectra using a version of the Fast Atmospheric
Signature Code 3 (FASCODE3) model which is for the retrieval of atmospheric trace gas profiles.
Error residuals between measured and fitted atmospheric transmission lie with in ±1.0% for $CO_2$
(6180-6380 cm$^{-1}$) and $O_2$ (7800-7940 cm$^{-1}$) respectively. During analysis period mean (standard
deviation, 1$\sigma$) $XCO_2$ was observed to be 385.24 ppm (4.22 ppm) with 1.0 % of daily variation.
Minimum and maximum averaged molecular column densities of $CO_2$ ($O_2$) are $6.15\times10^{21}$
($3.3\times10^{24}$) molecules/cm$^2$ and $8.06\times10^{21}$ ($4.72\times10^{24}$) molecules/cm$^2$ respectively. Obtained an
average high signal to noise ratio (SNR) of 833 and 625 for NIR and MIR spectra, respectively.
Keywords: Column $CO_2$, Vertical profile, FTIR, FASCODE3, Trace gases
**1. Introduction**
Greenhouse and other trace gases such as carbon dioxide ($CO_2$), methane ($CH_4$), ozone ($O_3$) and
nitrous oxide ($N_2O$) play a vital role in controlling the climate system of the lower atmosphere
(Stocker et al., 2013; Smedley et al., 2015; Sreenivas et al., 2016). $CO_2$ in the atmosphere is the
most important contributor to positive radiative forcing that increases the greenhouse effect
(Forster et al., 2007). It has increased about 40% from the year 1750 to 2011 with the level of
atmospheric abundance of $CO_2$ was 390.5 ppm (390.3 to 390.7) in the year 2011 (Stocker et al.,
2013). To understand better and manage $CO_2$ emissions, estimates of source and sink strengths
are required by the integrated approach of in-situ, remote sensing and model simulation. Currently
the information about atmospheric $CO_2$ is mainly inferred from in situ (Warneke et al., 2005; Petri
et al., 2012) and remote sensing technology.
Rayner and O'Brien (2001) have shown that space-based column $CO_2$ can substantially improve
understanding of surface fluxes only if they have accuracy and precision of 1-2 ppm with good
spatial and temporal coverage. National Aeronautics and Space Administration (NASA) launched
a dedicated Orbiting Carbon Observatory-2 (OCO-2) satellite in 2014 to measure column $CO_2$
(http://oco.jpl.nasa.gov/). The OCO-2 instrument aimed to measure $XCO_2$ with a precision better
than 0.3 % on spatial scales (<100 km). Air-borne and satellite based measurements of such
parameters are subjected to uncertainties associated with the constraints related to the retrieval
techniques and limitations inherent to the sensor (A.P. Cracknell & C.A. Varotsos 2014). Thus,
direct measured column measurements are potentially a decisive input for atmospheric $CO_2$
inversion because of lower impact from errors in modeled vertical convection (Warneke et al.,
2005; Kobayashi et al., 2010). Column measurements are especially important in the tropics, as





convection is consistently strong and as a result flux signals are only weakly seen in surface measurements (Gloor et al., 2000).

National Remote Sensing Center (NRSC) of Indian Space Research Organization (ISRO) has established a dedicated Atmospheric Sciences Lab (ASL) to record, monitor and analyzes the greenhouse and other trace gases along with radiation measurements towards understanding the impact of atmospheric processes and assess the air quality. A high resolution (maximum $\Delta v$=0.0035 cm$^{-1}$; optical path depth (OPD) of 257 cm) FTIR spectrometer for measuring atmospheric trace gases and GHGs has been installed and currently operational on clear sky days at ASL, Shadnagar since March 2014. We hope these measurements provide highly reliable and accurate standard data sets over a long period. Additionally, they provide complementary information to the satellite measurements such as diurnal variations.

In the present study, NIR and MIR spectra have been utilized to retrieve column averaged and vertical profile of $CO_2$ using LBLRTA and radiative inverse model (FASCODE3).

## 2. Data measurement and analysis

High resolution ($\Delta v$=0.01 cm$^{-1}$; OPD=90 cm) Sun spectra have been obtained using ground based FTIR spectrometer over Shadnagar region (Latitude=17.03° N, Longitude=78.21° E) of India have been evaluated for a week. An IFS 125M FTIR housed at ASL is equipped with set of beam splitters and detectors such as Potassium Bromide (KBr) and Calcium fluoride (CaF$_2$) beam splitters and Mercury Cadmium Telluride (MCT) and Indium antimonide (InSb) detectors cooled with liquid nitrogen (LN$_2$). Details are summarized in table 1. In the present study, the measured spectra using InSb detector to understand the typical columnar concentrations of $CO_2$ by implementing the basic LBLRTA at 6100 cm$^{-1}$-6400 cm$^{-1}$ spectral range. Also attempted to retrieve $CO_2$ vertical profile information in NIR and MIR spectral range using FASCODE3 model (Notholt 1994). The FASCODE3 (Smith et al., 1978; Wang et al., 1996) coupled with an inversion module is based on the optimal estimation method of Rodgers (1976), provides error analysis tools necessary to determine the information content of the retrievals. The reader is referred to detailed retrieval methods and their error analysis explained in Miller et al. (1999). Observations covering the spectral range from 6000 cm$^{-1}$ to 8000 cm$^{-1}$ were used to retrieve the column averaged dry mole fraction (DMF) of $CO_2$ and vmr of $O_2$ (Wallace and Livingston, 1990 and Yang et al., 2002).

Implemented LBLRTA for the analysis of solar spectra absorption features to retrieve the columnar abundance of $CO_2$ in the atmosphere. In the present study, the absorption bands for $CO_2$ at 6100 cm$^{-1}$-6400 cm$^{-1}$ and for $O_2$ at 7800 cm$^{-1}$-7960 cm$^{-1}$ have been used. Retrieval includes various options such as scaling of *a priori* profiles of pressure (P), temperature (T) and volume mixing ratios (vmr). In this study, we have used *a priori* profiles (vmr) simulated over Izana (Tenerife, 28.3° N, 16.5° W) by National Center for Atmospheric Research/Whole Atmosphere Community Climate Model (NCAR/WACCM). Typical climatological profile of $CO_2$ and (P, T) obtained for 44 levels (~120 km) from NCAR/WACCM and analysis provided by NASA Goddard Space Flight Center (GSFC, science@hyperion.gsfc.nasa.gov). The $CO_2$ and $O_2$ spectral parameters were obtained from High Resolution Transmission (HITRAN) 2012 database



(Rothman et al., 2013). Figure 1 provides atmospheric signal measured over Shadnagar with
different combination of detectors and beam splitters.
Spectra were recorded at 0.01 cm$^{-1}$ resolution. It is possible to derive zenith column densities of
$CO_2$, $CH_4$, $N_2O$, CO, $O_3$, $C_2H_6$ and HF. The present study reports retrievals of $CO_2$ and $O_2$ gases
for one week during 08[th], 10[th], 15[th], 16[th], 21[st] and 23[rd] March 2016.

### 3.  Results and discussion

The measured spectra were analyzed at various times throughout the day particularly solar zenith
angle (SZA) at around nadir view (±75.0°) to obtain the atmospheric signal over the study region.
The spectra are analyzed are generally co-additions of 2-4 individual spectra, each taking 5 minutes
to acquire. In our analysis, an average SZA taken to be 75°.0.  Figure 2 shows a sample (21[st] March
2016) spectral analysis in which two spectral bands of $CO_2$ and $O_2$ were fitted against measured.
FASCODE3 model has been used to fit the $CO_2$ and $O_2$ spectra against measured transmittance
spectra. It computes spectral transmittance, radiance and optical depth for a given spectral range.
The spectral line parameters are based on the latest HITRAN line list (Rothman et al., 2013). Heart
of the FASCODE3 is a line-by-line calculation which computes atmospheric transmittance at very
high spectral resolution. Summary of the retrievals and model inputs with SNRs of spectral
windows provided in table 2.
In our analysis, we used central wave numbers ($\nu_c$) for $CO_2$ and $O_2$ are 6348 cm$^{-1}$ and 7808 cm$^{-1}$
respectively. It has an advantage of being collected using the same detector and the ratio of $CO_2/O_2$
will mostly cancel out the systematic effects such as instrument line shape (ILS). In figure 2b and
2d residual (measured-calculated) errors were shown for $CO_2$ and $O_2$ that lie within ±1.0%
respectively. The time series column-averaged concentration of $CO_2$ were shown in figure 3. These
limited spectra particularly selected to obtain vertical column abundance at near nadir view over
Shadnagar region. Thus, these spectra were obtained during 11:30 LT to 12:30 LT for 5 min
interval where SZA is 75.0°. The ratio of $CO_2$ and $O_2$ column amounts, scaled by the standard
atmospheric $O_2$ fraction have been computed as

$$\text{XCO2}_{\text{DMF}}(\text{ppm}) = 0.2095 \times \left[ \frac{\text{Column of X}_{\text{CO2}}}{\text{Column O}_2} \right] - - - (1)$$


Time series observations were averaged within the selected time interval which may probably
reduce the thin clouds impact on absorption features and instrumental effects such as line shape.
The column-averaged $CO_2$ was observed to be minimum (maximum) of 381.0 ppm (390.0 ppm).
Minimum and maximum averaged molecular column densities of $CO_2$ ($O_2$) are $6.15 \times 10^{21}$
($3.3 \times 10^{24}$) molecules/cm$^2$ and $8.06 \times 10^{21}$ ($4.72 \times 10^{24}$) molecules/cm$^2$ respectively. Daily means of
the standard deviations (1σ) are 3.04 ppm, 3.58 ppm, 3.40 ppm, 0.74 ppm, 0.84 ppm and 3.12 ppm
respectively.


In the present study, using FASCODE3 we retrieved $CO_2$ concentration in the vertical profile for
one day (21st April 2015 and 23rd March 2016) in two consecutive years as shown in figure 4.
These retrievals are critically dependent on atmospheric pressure and temperature profiles. Present
study used NIR (6100-6400 cm$^{-1}$) and MIR (660-700 cm$^{-1}$) spectral windows for sensitivity
comparison for two different days in different years. Both the profiles are likely to be same with
the varied temperature input. The mean standard deviations (1 σ) for NIR and MIR are 0.07 ppm
and 0.08 ppm respectively. Since the temperature and pressure profiles are essential to the retrieval,
a priori profile information of (P, T) obtained from reanalysis data provided by NASA-GSFC
(science@hyperion.gsfc.nasa.gov). The criteria for choosing these bands also include the relative
spectral line intensities, the number of lines and minimization of molecular species interferences
on main species. FASCODE3 also computes the strength of line intensities in the retrieval window
and the total number of lines (shown in table 3).
For the spectral band in the NIR region (6100-6400 cm$^{-1}$), spectral line lists are 14,247 which are
less compared to that of this band in the MIR region (660-670 cm$^{-1}$) of 44,065 lines. Fortunately,
these two spectral regions are less influenced by other molecular species as shown in table 3.
**4. Conclusion**
This work presents the column-averaged concentration and an attempt made to retrieve vertical
profile of $CO_2$ concentration using a line-by-line radiative transfer algorithm and FASCODE3 over
Shadnagar region of India. Six day direct solar radiation spectra were utilized with high spectral
resolution of 0.01 cm$^{-1}$ recorded by ground-based FTIR spectrometer. Measured transmittance
spectra compared against model computed transmittance spectra and found to be good agreement
with high SNR. The mean residual lie within ± 1.0% for $CO_2$ and $O_2$ spectral windows. During the
analysis period mean (standard deviation, 1σ) $XCO_2$ was observed to be 385.24 ppm (4.22 ppm)
with 1.0 % of variation in selected SZA period. This precision has to be improved by using near
real time meteorological information acquired by INSAT-3D. Present study used NIR (6100-6400
cm$^{-1}$) and MIR (660-700 cm$^{-1}$) spectral windows to compare retrieved vertical profile of $CO_2$
during two different days. The mean standard deviations of retrieved $CO_2$ (1 σ) for NIR and MIR
bands are 0.07 ppm and 0.08 ppm respectively. To understand the spectral dependencies and
resolution for trace gas retrievals, further analysis required using FASCODE3 and that would form
the future work. In addition, attempts would be made to use near real time temperature and pressure
profiles from satellite data to improve the retrieval accuracy.
**Acknowledgement**
The authors sincerely thank colleagues of Atmospheric and Climate Sciences Group (ACSG) of
Earth and Climate Science Area (ECSA) for their immense support in collecting the data. We
would like to acknowledge Dr. J. Hannigan, NCAR, Boulder, CO and Dr. Frank Hase, project
manager Collaborative Carbon Column Observing Network (COCCON) for sharing a priori
climatology data and their valuable suggestions. The authors specially thank our colleague Mr.
Biswadip Gharai for his valuable suggestions and support during this work.





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



**Tables**
Table 1 Measurement details using FTIR 125M (Bruker make)

| Detector | Beam Splitter | Spectral Range (cm$^{-1}$) | Resolution ($\Delta v$) cm$^{-1}$ | OPD (cm) | SNR | Noise (rms) |
|---|---|---|---|---|---|---|
| **MCT** | KBr | 600-4800 | 0.01 | 90 | 625 | 0.16 |
| **InSb** | CaF$_2$ | 1000-11000 | 0.01 | 90 | 833 | 0.12 |


Table 2 Model inputs and retrieved column averaged $CO_2/O_2$ windows

| Detector | Beam splitter | Retrieval window (cm$^{-1}$) | Central wavenumber ($v_c$, cm$^{-1}$) | Species | SNR | Noise(rms) |
|---|---|---|---|---|---|---|
| MCT | KBr | 660-700 | 668 | $CO_2$ profile | 1111 | 0.09 |
| InSb | CaF$_2$ | 6100-6400/7800-7960 | 6348/7808 | $XCO_2/O_2$ | 2941/2564 | 0.03/0.04 |
| Inputs-FASCODE3 | | | | | | |
| Spectra library (HITRAN2012) | SZA | Observed height from mean sea level (km) | Modified Atmospheric (tropical) model | Retrieval height (km) | Mean mixing ratio of $CO_2$ (assumed 390 ppm, Stocker et al. 2013) | |


Table 3 FASCODE3 computed line strength in NIR (6100 cm$^{-1}$-6400 cm$^{-1}$) and MIR (660 cm$^{-1}$-700 cm$^{-1}$)
spectral region

| Molecule | No.of Lines (NIR/MIR) | Line Strengths (sum) (NIR/MIR) |
|---|---|---|
| $H_2O$ | 1543/937 | 2.61E-27/2.97E-23 |
| $CO_2$ | 14247/44065 | 1.55E-25/1.377E-20 |
| $N_2O$ | 236/1257 | 3.90E-26/4.405E-22 |
| CO | 677/0 | 6.84E-26/0.0 |
| $O_2$ | 91/0 | 5.04E-33/0.0 |

274





275    **Figures**



Figure 1 FTIR Measured Solar Spectra in the NIR and MIR spectral regions

Figure 2 NIR spectra measured on 21st March 2016. a) Red lines indicate measured transmittance and black lines represent fitted transmittance for $CO_2$ b) residual error for $CO_2$ c) Transmittance fitted versus measured in $O_2$ spectral range d) residual error of $O_2$





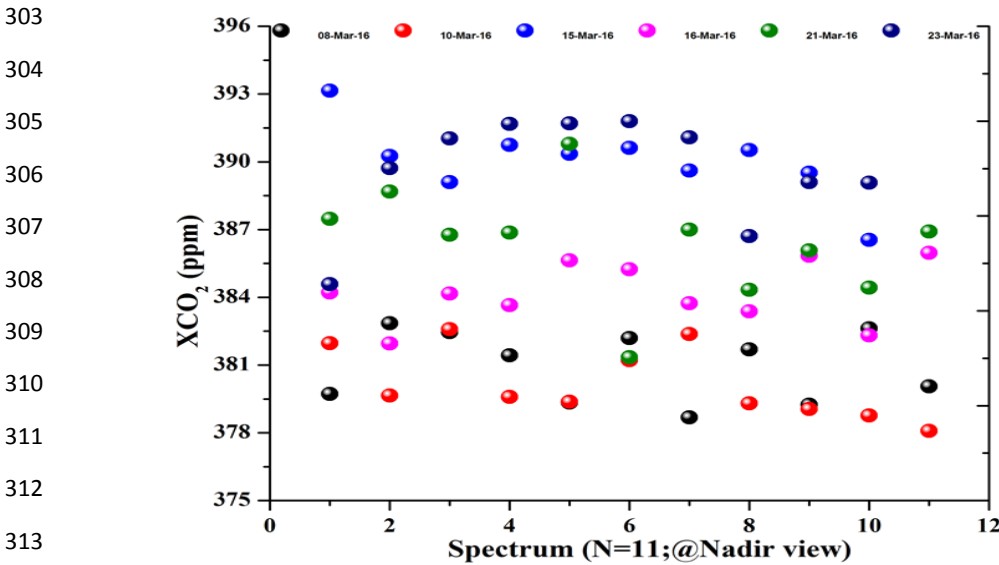

Figure 3 Time series column-averaged volume mixing ratio of $CO_2$ ($XCO_2$) over Shadnagar region

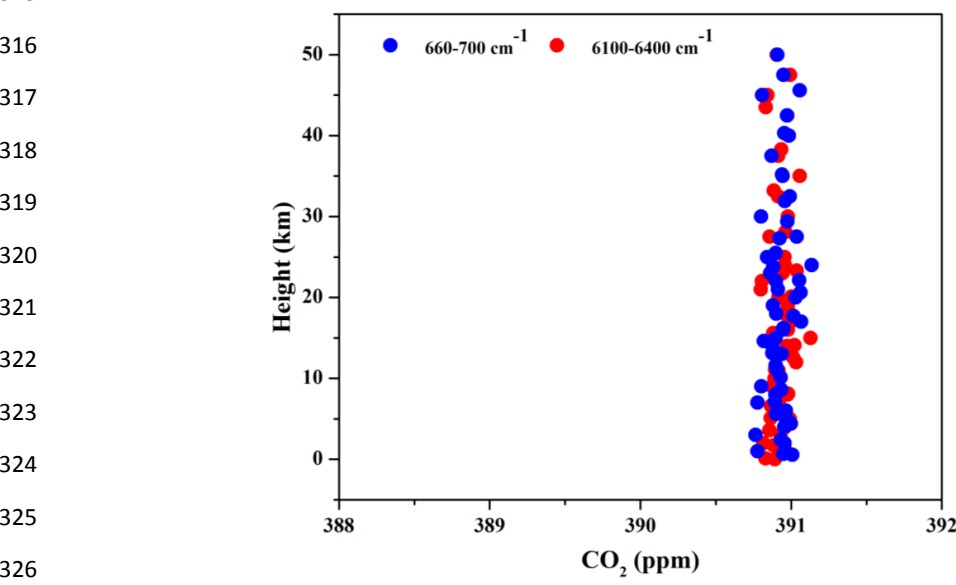

Figure 4 Volume mixing ratio of $CO_2$ retrieved on 21st April 2015 and 23rd March 2016