# Peer review of "Atmospheric CO₂ retrieval from ground based FTIR spectrometer over Shadnagar, India."

_Atmospheric Measurement Techniques, 2016_

## Referee Comment (RC1) · Anonymous Referee #1 · 28 Jun 2016

This paper describes ground based solar Fourier transform spectroscopy remote sensing measurements of total column atmospheric CO2 from Shadnagar, India. Similar measurements as part of the global TCCON network have become the de facto gold standard for validation of satellite-based measuremnts by GOSAT, OCO-2 and other instruments, and are used independently in model inversions to elucidate the global carbon cycle. It would be very desirable to complement the TCCON network in the region of the Indian sub-continent, as there is currently no coverage there.

Unfortunately, the work described in this paper falls far short of the accuracy and precision required and already demonstrated in the TCCON and NDACC networks, as published in several papers and the CDIAC data base by Wunch et al. and many others. Further, the authors make no acknowledgement of, or reference to, the large existing body of closely-related work in TCCON - this is inexcusable in such a publication, there are over 100 publications listed on the public TCCON wiki. The authors appear to have done this work with little knowledge or recognition of the existing state of the science. The data presented are a small snapshot of demonstrably inferior accuracy, adding little or nothing to either the advancement of techniques or the body of geophysical data. I therefore recommend that the paper be rejected.

I provide an indication of the major inadequacies in the work which would have to be addressed before resubmission for publication.

1. The demonstrated precision and daily variation of XCO2 is $\sim$ 4ppm, 5-10 times worse than that achieved in TCCON. The total column amounts around 390 ppm appear to be biased 5-10 ppm low based on the calibrated TCCON network results. To be of value for current satellite validation and model applications, the accuracy and precision needs to be improved by an order of magnitude.

2. The poor accuracy and precision appear to be due to both the actual collected spectra and the retrieval method used. The spectra shown in Fig 1 show very bad saturation, especially in the MIR region. It is well known in TCCON that the InSb detecor used here also saturates unless bandpass filters are used to restrict the photon flux. InSb is inferior to InGaAs detectors. Saturation will directly affect the accuracy of retrieved total column amounts. These aspects of the measurements are all described in the available published TCCON and NDACC literature but appear to have been ignored by the authors.

3. The retrieval method based on FASCODE3 is inadequately described - there is the perception that it is used as a "black box". Tne residuals displayed in Figure 2 show clearly that the forward model is not adequately fitting the measured spectra. The residuals are several times larger than those achieved with GFIT in the TCCON network or with other codes such as SFIT2/SFIT4 or PROFITT and indicate poor lineshape and position matching. However no details of the model are provided in the paper. No mention is made of how the solar spectrum is included in the forward model. Line

119 implies that only an average solar zenith angle around 75 degrees is used for all spectra - if true this is a major source of potential error and inadequate for the accuracy required for these measurements to be useful. Finally, the method for profile retrieval is not explained at all.

---

## Referee Comment (RC2) · Anonymous Referee #2 · 19 Jul 2016

The mansuscript "Atmospheric CO$_2$ retrieval from ground based FTIR spectrometer over Shadnagar, India" by Mahesh et al. describes – to my knowledge – the first ground based column-averaged dry-air mole fraction observations of CO$_2$ (aka XCO$_2$) over India. As such, these measurements could be very interesting if carried out over a longer time period. However, the current manuscript unfortunately lacks in a number of technical and scientific aspects.

**Major comments**

What I find most strange is that the authors basically repeat the approach of the Total Carbon Column Observing Network (TCCON): they use the same principal technique (Fourier-transform spectroscopy) to observe the same species in the same spectral regions. However, there is no mention whatsoever of TCCON nor are any of the dozens

of TCCON-related articles cited. Note that most of these articles appeared in open-access journals and are therefore easily available. So the mansuscript has clearly failed in summarizing the state-of-the-art.

Because they are not aware of the state-of-the-art, the authors fall into several traps that could have been avoided. They seem to use their own code for calculating spectra, radiative transfer and inversion. However, the referenced FASCODE 3 is very old (10+ years) and certainly not state of the art any more. The mentioned LBLRTA code is not referenced at all. While there is nothing wrong about using a different code, I would at least expect a comparison with some well-established code (GFIT, SFIT, PROFFIT). I fear that the authors have tried to reinvent the wheel here.

I am also not sure if the authors are aware of the very high precision and accuracy requirements for $XCO_2$ retrievals. The expected signals in the total column are in the order of only 1–3 ppm. To make a meaningful contribution, any new observations should have a precision of better than 0.25–0.3% ($\pm1$ ppm). That is a very demanding goal and it takes a number of preconditions to reach. The first is a very stable and well-aligned FTIR spectrometer. The TCCON community does not consider the Bruker IFS125M suitable for this purpose because of its lower stability compared to the Bruker IFS125HR and the difficulties of aligning the interferometer to the necessary modulation efficiency. If the authors think differently, they should prove the stability of their setup, e.g. by providing more information about their instrumental line shape or by showing Allan variance measurements. Another precondition for reaching this level of accuracy is the combination of retrieval algorithm and spectral line data. The combination of the old FASCODE3 and the unknown LBLRTA is not convincing without additional validation. And the spectral data of HITRAN 2012 is known to be not accurate enough for this purpose. Therefore, TCCON observations have to be calibrated vs. aircraft in-situ measurements (e.g. Messerschmidt et al., Atmos. Chem. Phys., 11, 10765-10777, doi:10.5194/acp-11-10765-2011, 2011).

One other issue is that the authors seem to use volume mixing ratio, mole fraction and

concentration synonymously. However, they are different quantities that are not to be mixed up. Mixing ratio and mole fraction are both dimensionless quantitles and the numerical difference may be small for many trace gases. It is however not small for gases like oxygen. If $n_i$ is the number of molecules of a certain species and $n$ is the total number of molecules in a given volume of (dry) air, then the mole fraction

$$x_i = \frac{n_i}{n} \tag{1}$$

while the mixing ratio

$$r_i = \frac{n_i}{n - n_i}. \tag{2}$$

For oxygen, the mole fraction $x_{O_2} = 0.2095$ whereas the mixing ratio $r_{O_2} = 0.2650$!

The combination of the factors mentioned above may also be the reason for the incredibly low $XCO_2$ values that the authors report in Fig. 3 and 4. The global $CO_2$ background already went above 400 ppm in 2014. To measure atmospheric $XCO_2$ values between 378 and 393 ppm anywhere in the world in 2016 would require an enormous $CO_2$ sink nearby. The day-to-day variability is also extremely high. It is much more likely that there is a strong bias in the order of 10–20 ppm in the results and the strong variability is the result of instrumental artifacts or an unstable retrieval.

**Minor comments**

The numerator in Eq. 1 should not be $XCO_2$.

**Conclusions**

I think $XCO_2$ observations in India would be very valuable and I hope that the following is not too discouraging. However, I do not think that the manuscript can be published as it is. The authors should contact the TCCON community and have someone look at their spectra and retrieval. I feel that the necessary changes go even beyond major revisions. My advice would be to re-submit the manuscript at some later time – possibly with more observations.

---

## Author Comment (AC1) · 2 Sep 2016

Referee #1 Dear Referee, Thank you very much for your valuable suggestions and comments. This certainly helped us to understand the importance of FTIR community. We are very sorry for not acknowledging the TCCON group and we admit our mistake. Presently due to some internal limitations, we were not able to be part of TCCON group and in near future, we shall ensure to be part of this group from India. This will certainly give mutual benefit to the scientific community. In the present study, we have implemented basic line-by-line radiative transfer algorithm (LBLRTA) based on the TCCON and NDACC literatures such as Rodgers (2000); Warneke et al., (2005); Wunch et al., (2010); Petri et al., 2012 and Messerschmidt et al., (2012). We agree that our retrievals are not meeting the TCCON standards. The reasons are assumed for the poor accuracy could be due to impact of clouds, varied instrumental line shape (ILS) and

solar zenith angle (SZA). Warneke et al. (2005) showed sensitivity of variable SZA on the precision and accuracy of the retrievals. FTIR 125M is using at our study site, due to AC coupling of the system, some of the spectra might impacted by passing clouds which probably introduce uncertainty in the retrievals. Since DC coupling method measures the variable transmission. These are our certain limitation for not achieving the high accuracy, which would be mostly negotiated with the support of TCCON in the near future.

1. The demonstrated precision and daily variation of XCO2 is $\sim$ 4ppm, 5-10 times worse than that achieved in TCCON. The total column amounts around 390 ppm appear to be biased 5-10 ppm low based on the calibrated TCCON network results. To be of value for current satellite validation and model applications, the accuracy and precision needs to be improved by an order of magnitude.

Ans: Thank you very much for your observations. We agree that the bias is at higher end when compared to TCCON sites where accuracy is below 0.2-0.5 % ($\sim$ 2 ppm). Retrievals of columnar observations up to 4 ppm (1.0 %) accuracy have been accomplished with IFS 125M FTIR in the present study while previous studies have reported the necessity for an accuracy of 1-2 ppm (0.2%). Due to aforementioned limitations, we could achieve an accuracy of 1 % ($\sim$4 ppm). This accuracy partly influenced by the less stable instrumental line shape (Hase et al., 2004; Warneke et al., 2010), an imperfect a priori information and clouds. Since the focus of the study is retrieve the columnar concentration of trace gases at best possible accuracy using LBLRTA which was implemented based on the information available from the TCCON literature.

2. The poor accuracy and precision appear to be due to both the actual collected spectra and the retrieval method used. The spectra shown in Fig 1 show very bad saturation, especially in the MIR region. It is well known in TCCON that the InSb detector used here also saturates unless bandpass filters are used to restrict the photon flux. InSb is inferior to InGaAs detectors. Saturation will directly affect the accuracy of retrieved total column amounts. These aspects of the measurements are all described in

the available published TCCON and NDACC literature but appear to have been ignored by the authors.

Ans: Kiel et al. (2016) showed clear difference between InSb and InGaAs detector recorded solar spectra. At our site in India, we used InSb detector and CaF2 beam splitter for NIR measurements and MCT and KBr beam splitter for MIR region. We agree that, without band pass filters detector gets saturated and reduce incoming photon flux. Our attempt to retrieve columnar concentration of CO2 was in NIR region. Due to uncorrected source brightness fluctuations and solar zenith angle influence, retrieval of XCO2 is not confirming to the TCCON standards. These impacts on retrievals accuracy and precision explained by many TCCON observations, which have been cited in the revised manuscript.

3. The retrieval method based on FASCODE3 is inadequately described - there is the perception that it is used as a "black box". The residuals displayed in Figure 2 show clearly that the forward model is not adequately fitting the measured spectra. The residuals are several times larger than those achieved with GFIT in the TCCON network or with other codes such as SFIT2/SFIT4 or PROFITT and indicate poor lineshape and position matching. However no details of the model are provided in the paper. No mention is made of how the solar spectrum is included in the forward model. Line 119 implies that only an average solar zenith angle around 75 degrees is used for all spectra - if true this is a major source of potential error and inadequate for the accuracy required for these measurements to be useful. Finally, the method for profile retrieval is not explained at all

Ans: We agree that the retrievals from SFIT2/SFIT4 or PROFITT and GFIT achieve TCCON standards, which can be able to incorporate DC-interferogram and correct the instrumental line shapes. In the present study also, we attempted basic LBLRTA which is a heart of those models. Figure 2 shows particularly in the case of O2, fitting was not adequate due to source brightness fluctuation and imperfect inputs to the standard model. These are some of the reasons for unexpected deviations. Below

flow chart shows the how column retrieval have been obtained from the measured solar spectra. In FASCODE3, we have modified (P,T) profile information obtained from GSFC, science@hyperion.gsfc.nasa.gov. Details of the FASCODE3 model and its applications are given by Smith et al., (1978); Notholt et al., (1994) and Wang et al., (1996). I am sorry for misrepresentation of SZA instead SZA are 75°-90° (11:30 LT-12:30 LT). Sensitivity of precision and accuracy of retrieved columnar CO2 also dominated by solar zenith angle. Attached figure shows the retrieval method which we implemented based TCCON literature.

Please also note the supplement to this comment:
http://www.atmos-meas-tech-discuss.net/amt-2016-177/amt-2016-177-AC1-
supplement.zip

- L-by-L radiative transfer algorithms accommodates various inputs such as standard atmospheric (P, T) profiles and volume mixing ratio (vmr)/a priori information of different gases.

- (P, T) profiles obtained from NCAR model simulated outputs

- NCAR/Whole Atmosphere Community Climate Model (WACCM) simulated Volume mixing ratios (vmr) profiles were utilized as a initial guess to LBLRTM

- Spectral line list information about each gas obtained from HITRAN spectroscopy data base

- Using HITRAN data base, we have calculated molecular absorption cross section which is proportional (Beer's law) to columnar density of a gas in the atmosphere.

- Solar absorption spectra and solar zenith angle information from IFS 125M spectrometer

- To minimize the retrieval uncertainty due to observational constraints, dry mole fraction was computed by taking atmospheric columnar concentration of $O_2$ molecule.

[Figure]

**Fig. 1.**